# Reversible Sorptive Preconcentration of Noble Metals Followed by FI-ICP-MS Determination

**DOI:** 10.3390/molecules27196746

**Published:** 2022-10-10

**Authors:** Yulia A. Maksimova, Alexander S. Dubenskiy, Lyudmila A. Pavlova, Ilya V. Shigapov, Dmitry M. Korshunov, Irina F. Seregina, Vadim A. Davankov, Mikhail A. Bolshov

**Affiliations:** 1Chemistry Department, Analytical Chemistry Division, Lomonosov Moscow State University, 1-3 Leninskie Gory, 119991 Moscow, Russia; 2Geological Institute of the Russian Academy of Sciences, Pyzhevsky lane 7, 119017 Moscow, Russia; 3A.N. Nesmeyanov Institute of Organoelement Compounds, Russian Academy of Sciences, 28 Vavilova Street, 119991 Moscow, Russia; 4Institute for Spectroscopy, Russian Academy of Sciences, 5 Fizicheskaya Street, Troitsk, 142190 Moscow, Russia

**Keywords:** noble metals, sorption, ICP-MS

## Abstract

In this paper, we propose the combined procedure of noble metal (NM) determination, including fire assay, acid digestion, and reversible dynamic sorptive preconcentration, followed by flow-injection ICP-MS. Reversible preconcentration of all NMs was carried out using micro-column packed new PVBC-VP sorbent and elution with a mixture of thiourea, potassium thiocyanate, and HCl, which recovers Pd, Ir, Pt, and Au by 95% and Ru, and Rh by 90%. The proposed procedure was approved using certified reference materials.

## 1. Introduction

Noble metals (NMs: Ru, Rh, Pd, Os, Ir, Pt, Au) are a very important part of modern industry. All branches of industry have used NMs directly or by implication, but natural NM sources are exhaustible. Therefore, it is economically attractive to estimate the degree of NM extraction from low-grade deposits. In addition, NMs are a tracer of anthropogenic activity [1,2,3] and a marker of various geological processes [4,5]. Due to all of these reasons, simultaneous determination of trace and ultra-trace NM contents in technological and especially in geological objects is an important analytical problem, the solution of which is impossible without preconcentration, even when using one of the most sensitive analytical methods: inductively coupled plasma mass spectrometry (ICP-MS).

Whilst also being eco-friendly, sorption is the easiest-to-use (especially in dynamic mode) method of preconcentration, and it can easily be scaled up for a large amount of samples. However, there are some common difficulties in sorptive preconcentration of noble metals: failure of group desorption of noble metals, influence of noble metal forms on sorptive degree, and low sorptive capacity. Usually, sorbents only have solutions for two of the three above problems. Sorbents for NM recovery may have different functional groups containing nitrogen atoms (N-containing) [5,6,7,8,9,10,11,12,13,14,15,16,17,18], sulfur atoms (S-containing) [19], N and S together [12,20,21], selenium atoms [22], phosphorus atoms [23], or oxygen atoms [24]. The N- and S-containing sorbents are the most common groups. S-containing sorbents for NM recovery are complex-forming resins; due to their structure, they have high capacity and are less sensitive to NM, but NM desorption from those sorbents is very low or impossible (see examples in Table 1). An N-containing sorbent may be as an ion-exchanger as well a complex-forming resin. N-containing complex-forming sorbents have the same problems as S-containing ones (see examples in Table 1).

N-containing ion-exchanger resins usually have lower capacity than complex-forming ones, but NMs may be desorbed totally or partially (see Table 2).

In the overwhelming majority of cases, sorbents with a pyridine group are complex forming due to the fact that the nitrogen atom is not quaternary because it usually does not have a substitute. Thus, N-containing sorbents are complexing resins or have low sorptive capacity in the case of ion-exchanger resins.

The simultaneous determination of NMs (except Os) after preconcentration by hyper-crosslinked polystyrene sorbent using ion-pair reagents [27] should be specifically mentioned. Unfortunately, this elegant technique turned out to be too time consuming for routine analyses of geological objects.

Magnetic sorbents—which have gained popularity for NM recovery—should be included in a separate category. A very informative review about magnetic sorbents is presented in articles [3,28]. Magnetic sorbent capacity is related to the structure and may amount from 100 μg g^−1^ to about 200 mg g^−1^. There are still no data about NM group sorption and recovery, but data about Pd, Pt, and Au recovery are very encouraging.

Therefore, the search for a new quaternary N-containing resin with good sorptive capacity for the group determination of very low NM content is still the problem at hand. The aim of our research to select a reversible sorptive system based on a new highly crosslinked copolymer containing pyridinium groups (with quaternary nitrogen atoms) for NM simultaneous determination. In addition, part of our research is focused on the spectroscopic investigations of the nature of the PVBC-VP resin interaction with the individual NM chlorocomplexes.

## 2. Results

### 2.1. Synthesis of PVBC-VP

PVBC-VP is a highly crosslinked copolymer containing one pyridinium group (1 PG) in repeating units; in this work, it was synthesized by the reaction of linear poly(4-vinylbenzylchloride) with 4-vinylpyridine [29] (see Figure 1). A strong dark cherry gel with a smooth surface was obtained and crushed to 315 μm.

### 2.2. FTIR of Blank PVBC-VP and after Sorption of [AuCl_4_]^−^

A FTIR spectrum of blank PVBC-VP was obtained (see Figure 2). There are typical bands of the pyridine ring (at ~1640, ~1570, ~1510, and ~1464 cm^−1^) in the FTIR spectrum, characterizing vibrations of the C=N, C=C, and C–H bonds and the skeletal vibrations of the pyridine ring in the copolymer containing N-alkylated poly(4-vinylpyridine). There is also a band at ~812 cm^−1^, characterizing the bending vibrations of the para-substituted pyridine ring. Note that the band at ~1640 cm^−1^ is the characteristic absorption band of the quaternized pyridinium group [30,31]. Other bands at about 3000 cm^−1^ and in the range 1600–1400 cm^−1^ belong to vibrations of aromatic or pyridinium rings entering into polymer structure. The broad band at ~3400 cm^−1^ is due to OH vibrations in the polymer spectrum because the sorbent was air-dried.

The FTIR spectrum of PVBC-VP after passing flow of [AuCl_4_]^−^ solution was also obtained (see Figure 3).

The new peak at 354 cm^−1^ present in the spectrum of polymer after Au sorption belongs to the Au–Cl bond, which is consistent with the [AuCl_4_]^−^ anion [32]. In addition, there are some differences between blank and after-sorption polymer spectra in the ranges 1400–1220 cm^−1^ and 1700–1500 cm^−1^.

### 2.3. Sorptive Conditions and Capacity of PVBC-VP

The different degrees of NM chlorocomplex sorption (S, %) from solutions with different HCl concentrations have been calculated using Equation (1) (see Table 3). These data were obtained in dynamic mode from model solutions with NM concentrations at a level below 100 μg L^−1^ and at the level of 1000 μg L^−1^. The model solution at a level below 100 μg L^−1^, prepared by digestion of SOMB-6 after fire assay, was used as a stock solution. At the level of 1000 μg L^−1^, single-element solutions for ICP-MS (for Pd, Os, Pt, and Au) and solutions prepared from salts (for Ru, Rh, and Ir) were used as stock solutions. All solutions were prepared by diluting stock solutions immediately before sorption experiments. All hypothesizes about the form of NM complexes in HCl solution were assumed based on well-known data from the literature [33,34].

The smallest range of HCl concentration for a quantitative extraction (more than 90%) was obtained for Rh and Ir complexes: 0.1–1.0 M HCl. Ru complexes were quantitatively extracted from a wider range: from 0.1 to 2.0 M HCl. The complexes of Pd, Os, and Pt have the same range of quantitative extraction: from 0.1 to 3.0 M HCl. Sorption degrees of those complexes from more concentrated acid solutions have not been obtained. The Au complex was quantitatively extracted from the widest range: from 0.1 to 6.0 M HCl.

The maximum preconcentration factor (PF_S_) of sorbent was determined for solutions containing each of the NMs below 100 μg L^−1^, which were passed through a 150 mm^3^ volume of sorbent. Calculated using Equation (3), the PF_S_ is about 330.

Sorption degree is influenced not only by HCl concentration but also by the form of NM chlorocomplex in solution. Pd, Os, Pt, and Au have more stable forms of complexes in the long term. Ru, Rh, and Ir change their form by dimerization, aquation, and hydroxylation, even when in concentrated HCl media for a long period of time [34]. There are data about sorption degrees for different Ru chlorocomplexes in Table 4. Analogous data for Rh and Ir were presented in our previous article [35].

The experimental sorptive capacity of PVBC-VP for all NMs was also determined exactly or approximately within the main working acid concentration range (0.2–1.0 M HCl) (see Table 5).

In addition, it was experimentally determined that the sorptive capacity of PVBC-VP for gold in 6.0 M HCl solution decreased by 9% compared to the sorptive capacity in the main working acid range.

### 2.4. Conditions of Recovery

In our previous article [35], some data on the recovery of NMs were mentioned, but quantitative desorption of all NMs was not achieved. All degrees of recovery (R, %) were calculated using Equation (2). Based on our early results [35] and assumption about the nature of interaction between NMs and PVBC-VP sorbent, a thiourea (Tu)–thiocyanate (Tcy)–HCl mixture was chosen for further investigation in more detail.

Firstly, the influence of the Tu:Tcy ratio in the eluting mixture on NM recovery degree was studied using isomolar series (see Figure 4).

Some synergetic effect was demonstrated for all NMs. A Tcy:Tu ratio in the range of (0.5–0.6):(0.5–0.4) was chosen as a more narrow range for investigation.

The obtained Au recovery in the Tcy:Tu isomolar series (see Figure 5) is in good agreement with the isomolar series of Tcy-Tu dissolution of gold [36].

The Tcy-Tu mixture’s synergetic effects are presented in Table 6.

As can be seen, the maximum recovery was obtained when the Tcy-Tu mixture was passed through the sorptive column.

In addition, the optimal ratio of C(HCl):C(Tu) was obtained and is equal to 1.6:1. The optimal mixture contents are equal to (0.5–0.6 M):(0.5–0.4 M) Tcy:Tu and 0.8–0.65 M HCl. A 10 mL volume of this solution recovers Pd, Ir, Pt, and Au by more than 95% and Ru, Rh, and Os by more than 90%. The maximum preconcentration factor (PF_R_) of the full reversible sorptive preconcentration procedure is equal to 5.

This mixture should not be introduced into ICP-MS directly. To avoid equipment damage and plasma suppression, the liquid concentrate of NMs was introduced into ICP-MS using a flow-injection system with a 50 μL dosing loop [37].

The limits of detection (3σ) of NMs in ng g^−1^ of rock or ore, using the full procedure (detailed description is in the Section 4.4), have been defined and are shown in Table 7.

When PVBC-VP sorbent is used to extract NMs from low-content solutions, the sorptive column withstands at least 10 complete cycles without reducing the degree of sorption (see Figure 6).

### 2.5. Validation of Full Preconcentration Procedure

The developed complex procedure of NM determination was verified using certified reference materials (CRM) such as SARM-7, GPt-6, and GPt-5 (see Table 8).

## 3. Discussion

### 3.1. Influence of PVBC-VP Sorbent Structure on the Sorption of NMs

Because of the very ramose 3D structure of PVBC-VP resin, it has pores with very different sizes and is well swelling. These factors make PVBC-VP sorptive centers accessible for octahedron NM chlorocomplexes as well as square planar complexes. However, PVBC-VP resin still is a sorbent sensitive to chlorocomplex forms, especially when containing aqua molecules and hydroxyl groups as a ligand. This drawback has been overcome using the unified procedure of Ni_x_S_y_ fire assay followed by acid digestion. Transformation of labile chlorocomplexes was avoided through the storage of digested samples in a concentrated HCl and dilution immediately before sorption.

### 3.2. Interaction between NM Chlorocomplexes and PVBC-VP

Based on previous results [35] and data in Table 3, we can come to a conclusion about differences in the nature of interaction of PVBC-VP sorbent with Ru, Rh, and Ir and Pd, Os, Pt, and Au. We had assumed that N^+^ from the pyridinium ring forms an outer-sphere complex with Pd, Os, Pt, and Au chlorocomplex anions. Gold was chosen for the following FTIR analysis for having the highest sorptive capacity on the 1PG sorbent.

A new peak at 354 cm^−1^ present in the spectrum of the polymer after Au sorption belongs to the Au–Cl bond, which is consistent with the [AuCl_4_]^−^ anion. In the case of monosubstituted square planar complex AuCl_3_–L, this peak would be shifted to the range of 380–360 cm^−1^, corresponding with the type of ligand (L) [32].

In addition, a difference between the blank polymer and the polymer with Au was demonstrated in the range of 1700–1200 m^−1^; this range includes deformation vibrations bands of pyridinium and benzene rings. This difference is too low for us to discuss intra-sphere complex forming, but not outer-sphere complexing. Nevertheless, based on previous results and the new FTIR data, stronger interactions between the polymer and Pd, Os, Pt, and Au than Ru, Rh, and Ir may be assumed.

Although the hypothesis about forming strong complexes between Au and PVBC-VP sorbent was not confirmed, differences in the mechanism of Pd-Pt-Au and Ru-Rh-Ir sorption were not refuted. A previous hypothesis [35] about recovering solutions as mixtures containing electrolytes, acid, and complexing reagents remains relevant.

### 3.3. Conditions of Sorption

The synthesized PVBC-VP sorbent has a successfully high capacity for extracting NMs from digesting solutions of real geological objects, with significant differences between various NMs containing, for example, gabbro or ultramafic tails, where NM contents can differ by 10–20 times or even more. The sorptive capacity of the PVBC-VP sorbent is higher than the capacity of typical ion-exchanger resin for the extraction of NMs (Table 2) and is on the same level as some complexing sorbents (Table 1).

The sensitivity of the PVBC-VP sorbent to Ru, Rh, and Ir forms (see Table 4) in solution was compensated by unified procedure of pretreatment and dilution of geological samples (detailed description of the full procedure is in the Section 4.4).

In addition, the high capacity of PVBC-VP sorbent for Pd, Pt, and Au (101, 140, and 240 mg g^−1^, respectively); a remarkably wide range of working HCl concentrations (0.1–6.0 M for Au); and experimental confirmation for Pd and Pt (0.1–3 M) make this sorbent very interesting for NM anthropogenic contamination removal.

### 3.4. Conditions of Quntitative Recovery

In our earlier paper [33], the hypothesis about optimizing eluent solution for quantitative NM recovery as a mixture of electrolytes, acid, and complexing reagents was proposed. There are thiourea and SCN^−^ as complex-forming reagents, HCl as an acid and electrolyte, and K^+^ as an electrolyte in the Tcy-Tu-HCl mixture. Specifically, K^+^ is optimal for the elution of Ir; the recovery degree of Ir using 2M KCl solution was equal to 100%, as demonstrated in our previous article [35]. As demonstrated in Table 6, using KCl instead of KSCN did not allow quantitative recovery of all NMs. Based on Figure 4 and Figure 5 and Table 6, it can be argued that there exists a synergistic effect of the Tcy-Tu-HCl system for NM elution. It can be assumed that thiourea and SCN^−^ together form hetero ligand complexes with NMs, and that K^+^ makes those complexes more stable. The process of forming hetero ligand complexes of Au with Tcy and Tu as ligands has been studied and used in hydrometallurgical experiments [37,39].

Recovery degrees of Pd, Ir, Pt, and Au of more than 95% and Ru, Rh, and Os of more than 90% have been demonstrated, using a mixture consisting of Tcy:Tu:HCl in the ratio (0.5–0.6 M):(0.5–0.4 M):(0.8–0.65 M).

All NMs have been quantitatively eluted from the PVBC-VP sorbent, which favorably sets this sorbent apart from most complexing resins (see Table 1). The elution of NMs from our sorbent does not require high temperatures or microwaving, in contrast to, for example, POLYORGS.

The PVBC-VP sorptive column is reusable due to good regeneration by Tu-Tcy-HCl solution, and it can be used for, at minimum, 10 big cycles successively without loss of recovery quality. This fact makes the use of the PVBC-VP sorbent more cost efficient.

Unfortunately, the obtained LODs for NMs (see Table 5) are as remarkable as in our previous work [25]: 2–15 ng g^−1^ for Tcy-Tu-HCl mixture against 0.2–3.0 ng g^−1^ for the ion-pare sorptive procedure due to a higher salt background from the Tcy-Tu-HCl mixture. Nonetheless, the Tcy-Tu-HCl procedure allows for the determination of all NMs.

### 3.5. Validation of the Procedure

The developed complex procedure of NM determination (Figure 6) was verified using some certified reference materials (see Table 8). The osmium determining results were lower than the certified ones because about 70–80% of Os escaped as OsO_4_ in the process of open acid digestion of the sample. Our preliminary experiment with synthetic Ni_x_S_y_ containing Os demonstrated that using a reflux condenser in the process of aqua regia digestion permits an increase in the residual content of Os from ~30% to ~60%. However, these results deserve additional thorough research and verification.

The bigger error for Pd and Pt determination in GPt-5 could be explained by their low content, approaching LODs.

## 4. Materials and Methods

### 4.1. Reagents

Concentrated nitric acid (65%, p.a. grade, Merck, Germany and puriss. spec. grade, Komponent Reaktiv, Moscow, Russia), concentrated hydrochloric acid (37%, puriss. spec. grade, Sigma Tek, Moscow, Russia), and concentrated perchloric acid (70%, for analysis, Merck, Darmstadt, Germany) were used.

Single-element stock standard solutions (Inorganic ventures, Christiansburg, VA, USA) of 100 μg L^−1^ Os or 1000 μg L^−1^ Ru, Pd, Pt, or Au and Rh or Ir solutions prepared from salts (RhCl_3_·xH_2_O or K_2_IrCl_6_ by Voikov chemical plant) were used for capacity experiments. The ICP-MS-68A-C multi-element standard solution (High-Purity Standards, North Charleston, SC, USA) containing 10 mg L^−1^ Ru, Rh, Pd, Os, Ir, Pt, and Au was used to prepare calibration solutions for ICP-MS determination. A single-element 1000 mg L^−1^ Indium standard solution (High-Purity Standards, North Charleston, SC, USA) was used for ICP-MS determination. All solutions were prepared using deionized water (18.2 MΩ cm, Millipore, Molsheim, France).

Carbonyl nickel powder (Ni content of at least 99.7%, Normetimpeks, Moscow, Russia), sulphur (puriss. spec. grade, LabTeh, Moscow, Russia), sodium carbonate (p. a. grade, MZHR, Moscow, Russia), sodium tetraborate decahydrate (borax), and silicon dioxide (silica) (all of p.a. grade, ReaHim, Moscow, Russia) were used for NiS fire assay of certified reference material (CRM) platinum ore SARM-7 (pyroxenite, MINTEK, Randburg, South Africa), GPt-5 (chromitite, IGGE, Beijing, China), and GPt-6 (peridotite, IGGE, Beijing, China).

The copper alloy SOMB-6 (SRM 7202–95, EZOCM JSC, Yekaterinburg, Russia) was used as a source of noble metals for the preparation of stock and model solutions, obtained after Ni_x_S_y_ fire assay procedure followed by dissolution in aqua regia (HCl:HNO_3_ as 3:1 *v*/*v*.).

Thiourea, potassium thiocyanate, and KClO_4_ (Sigma Aldrich) and NaCl, KCl, NaClO_4,_ Mg_2_ClO_4_, and NH_4_ClO_4_ (p.a. grade, ReaHim, Moscow, Russia) were used for the preparation of desorption solutions.

Potassium bromide (>99.999% pure, Pike) was used for obtaining FTIR spectra.

### 4.2. Sorbent

The one pyridinium group (1PG) sorbent, named PVBC-VP, was synthesized by Davankov’s group in the Laboratory “Stereochemistry of Sorption Processes”, INEOS RAS, Moscow, Russia.

The PVBC-VP was prepared by the amination of linear poly(4-vinylbenzylchloride) with 4-vinylpyridine. 1PG has the most (about 75–77%) repeating units of the three-dimensional polymeric network. The final product presents as a highly crosslinked copolymer because the vinyl groups of 4-vinylpyridine undergo spontaneous ionic polymerization (see Figure 1 in Section 2.1). More details about the synthesis can be found in [29].

PVBC-VP is supposedly a strongly basic anion-exchange resin. The sorbent is well swelling in any aqueous or polar medium (4 g of water per g of polymer), but shrinks on drying and then displays a specific surface area of not higher than 40 m^2^ g^−1^ by argon desorption single-point BET technique. The particle size is about 315 μm. The exchange capacity of the 1PG polymer amounts to 3 meq g^−1^ in terms of Cl^−^.

### 4.3. Equipment

Fusion of the certified reference materials and copper alloy was performed in an alundum crucible in a muffle furnace: SNOL 10/11V (Labtest, Moscow, Russia).

A Plexiglas column of 3 mm i.d. and 20 mm bed depth was packed with the sorbent (bed volume of sorbent: about 150 mm^3^). Peristaltic pump ISMATEC REGLO Analog MS-4/12 (Vernon Hills Cole-Parmer US, IL, USA) and hosepipes TYGON (Saint-Gobain Performance Plastics, Paris, France) were used for solution passing through a sorptive column. 

NM determination were carried out with a quadrupole ICP-MS spectrometer, Agilent 7500c (Agilent Technologies, Tokyo, Japan), in time resolved analysis (TRA) mode. Sample introduction to the mass spectrometer was carried out using a flow-injection system consisting of a 6-port manual injector Rheodyne 9740 (Rheodyne LLC, Rohnert Park, CA, USA), a 50 μL PEEK sample loop (Agilent Technologies, Tokyo, Japan), and an HPLC pump (Series 1, Akvilon, Moscow, Russia). The 1% HNO_3_ solution was used as a mobile phase. The data were acquired and processed with the ICP-MS ChemStation software package (version G1834B, Agilent Technologies). The following isotopes were used: ^99^Ru, ^101^Ru, ^102^Ru, ^103^Rh, ^108^Pd, ^188^Os, ^189^Os, ^190^Os, ^193^Ir, ^195^Pt, and ^197^Au as analytes and ^115^In as the internal standard to account for the matrix effect.

The FTIR spectrum of PVBCh-VP was obtained using a Vertex 80v spectrometer. The spectra were recorded in a vacuum. Device operation mode: 64 scans, aperture slit: 8 mm, in the range 4000–400 cm^−1^ at 4 cm^−1^ resolution. For spectra processing, we used OMNIC software.

### 4.4. Sample Preparation

The full scheme of the determining procedure from sample pretreatment to ICP-MS is shown on Figure 7.

The procedure of Ni_x_S_y_ fire assay and following acid digestion was described in detail in our previous work [35]. Here is the sample digestion procedure in brief: CRM was melted with Ni powder, S powder, SiO_2_, Na_2_CO_3_, and borax under 1050 °C for 2.5 h in total. Obtained NM-containing Ni_x_S_y_ –matte was crushed and digested in aqua regia in an open system or under reflux condenser, and then it was converted to concentrated HCl.

The procedure of sorptive preconcentration in dynamic and static mode was also described in detail in our previous article [35]. Briefly, NM-containing 0,1–3 M HCl solutions were passed through a sorptive column in the forward direction at a rate of 2.0 mL min^−1^ by peristaltic pump. Waste solutions were collected and analyzed by ICP-MS. Eluting solutions were passed through a sorptive column in the inverse direction; the volume rate was 1.0 mL min^−1^ by peristaltic pump. Concentrates were collected and analyzed by ICP-MS.

The sorptive capacity for the inert complex ions of NMs (for example, [AuCl_4_]^−^) was obtained in batch conditions, but full sorptive capacity for more labile complex ions (for example, [RhCl_6_]^3−^) was determined in dynamic conditions. In both cases, the mass of a sorbent was equal to the mean of 20 mg and the volume of a solution was equivalent to 50 mL.

The degree of sorption (*S*, %) and the recovery (*R*, %) were used for estimating sorptive and elution efficiencies, respectively, and were calculated by the following equations:(1)$$S=100-\frac{m_{w}}{m_{o}}\cdot 100$$
(2)$$R=\frac{m_{el}}{m_{o}}\cdot 100$$
where *m_o_* is the initial content of noble metals in the model solution (ng); *m_w_* is the content of noble metals in the waste solution (ng); and *m_el_* is the content of noble metals in the concentrate (ng).

Sample preparation for obtaining FTIR spectra: 2 mg powder of ground sorbent was mixed with 200 mg KBr and pressed. All spectra have been normalized by the standard deviation within a sample.

The preconcentration factors (*PF*) were calculated for sorption and for full reversible sorptive preconcentration by same the following equation:(3)$$PF=\frac{V_{initial}}{V_{final}}$$
where *V_initial_* is the volume of initial solution of noble metals before passing through column (mL); *V_final_* is the volume of sorbent in column (0.150 mL) in case of calculating *PF* of sorption, and it is the volume of final recovery solution (mL) in case of calculating *PF* of full reversible sorptive preconcentration procedure.

## 5. Conclusions

The combined procedure of NMs determination, including Ni_x_S_y_ fire assay of initial sample, acid digestion, and reversible dynamic sorptive preconcentration followed by flow injection ICP-MS analysis, was proposed. The new PVBC-VP sorbent, containing one pyridinium group in repeating unit, placed in a 150 μL column, was used for preconcentration of all NMs.

This sorbent has good capacity for Pd, Pt, and Au equal 101, 140, and 240 mg g^−1^, respectively, and a remarkable range of working HCl concentrations equal to 0.1–6.0 M for Au and a minimum of 0.1–3.0 M for Pd and Pt. The ranges of working HCl concentrations for Ru and for Rh and Ir are equal to 0.1–2.0 M and 0.1–1.0 M HCl, respectively.

Based on the spectroscopic data (FTIR spectra are presented in this text and our previous UV- visible spectra) and previous desorption experiments with individual eluent solutions, an efficient mixed eluent for the quantitative recovery of analytes was proposed. Recovery degrees of Pd, Ir, Pt, and Au of more than 95% and Ru, Rh, and Os of more than 90% have been demonstrated, using a mixture consisting of Tcy:Tu:HCl in the ratio (0.5–0.6 M):(0.5–0.4 M):(0.8–0.65 M). The PVBC-VP sorptive column is reusable due to good regeneration by the Tu-Tcy-HCl solution and can be reused for a minimum 10 big cycles successively without loss of recovery quality.

The LODs for NMs, determining by proposed complex procedure of NMs, were calculated and equal to 2–15 ng g^−1^ in initial sample. The proposed complex procedure of all NMs was approved using CRM.

## Figures and Tables

**Figure 1 molecules-27-06746-f001:**
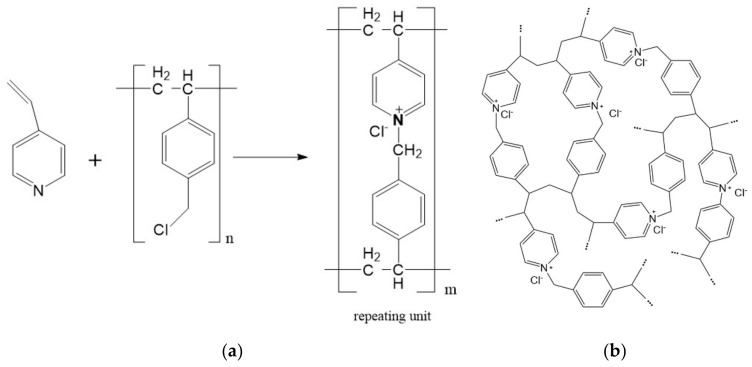
Scheme of synthesis and structure PVBC-VP: (**a**) scheme of synthesis and structure of repeating unit of PVBC-VP; (**b**) visualization of crosslinked net structure of PVBC-VP (where “⋯” symbolizes the continuation of the polymer chain).

**Figure 2 molecules-27-06746-f002:**
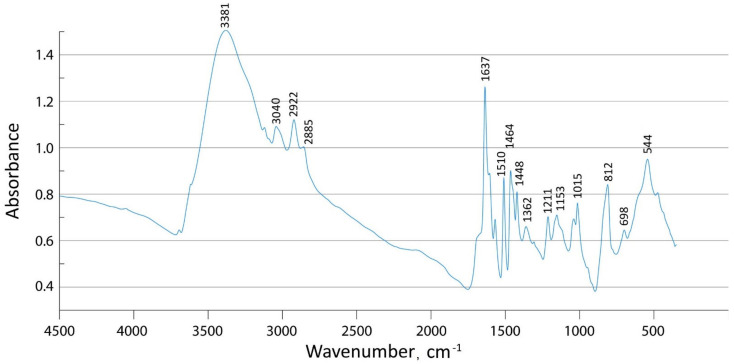
FTIR spectrum of PVBC-VP.

**Figure 3 molecules-27-06746-f003:**
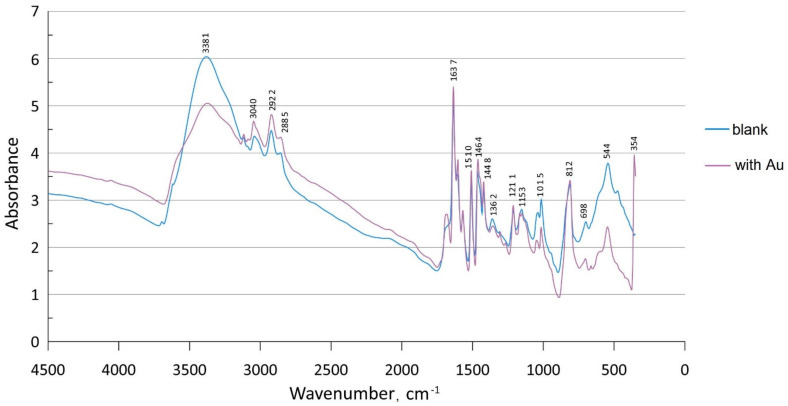
FTIR spectra of blank PVBC-VP (blue curve) and after sorption of [AuCl_4_]^−^ (purple curve).

**Figure 4 molecules-27-06746-f004:**
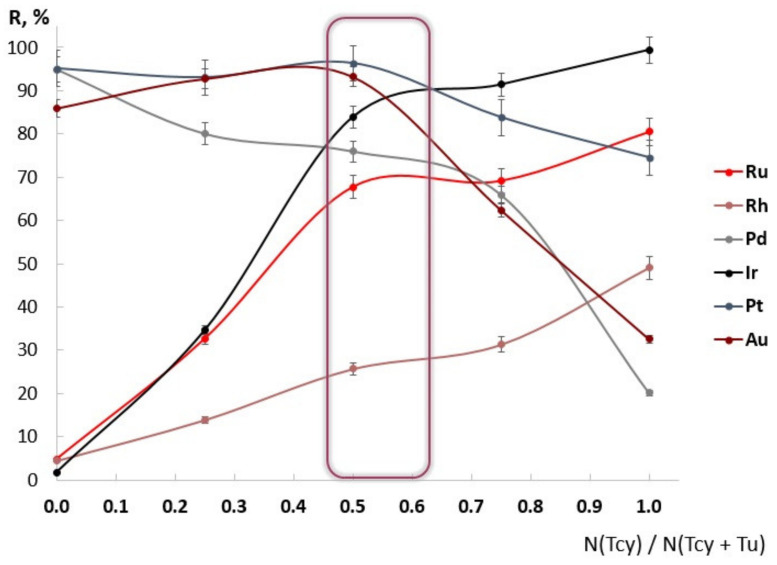
The isomolar series of desorption using thiourea–thiocyanate mixture in 0.2 M HCl (*n* = 3, *p* = 0.95). The sum of concentrations of thiourea and thiocyanate was equal to 0.5 M. Sorption from SOMB-6 solution.

**Figure 5 molecules-27-06746-f005:**
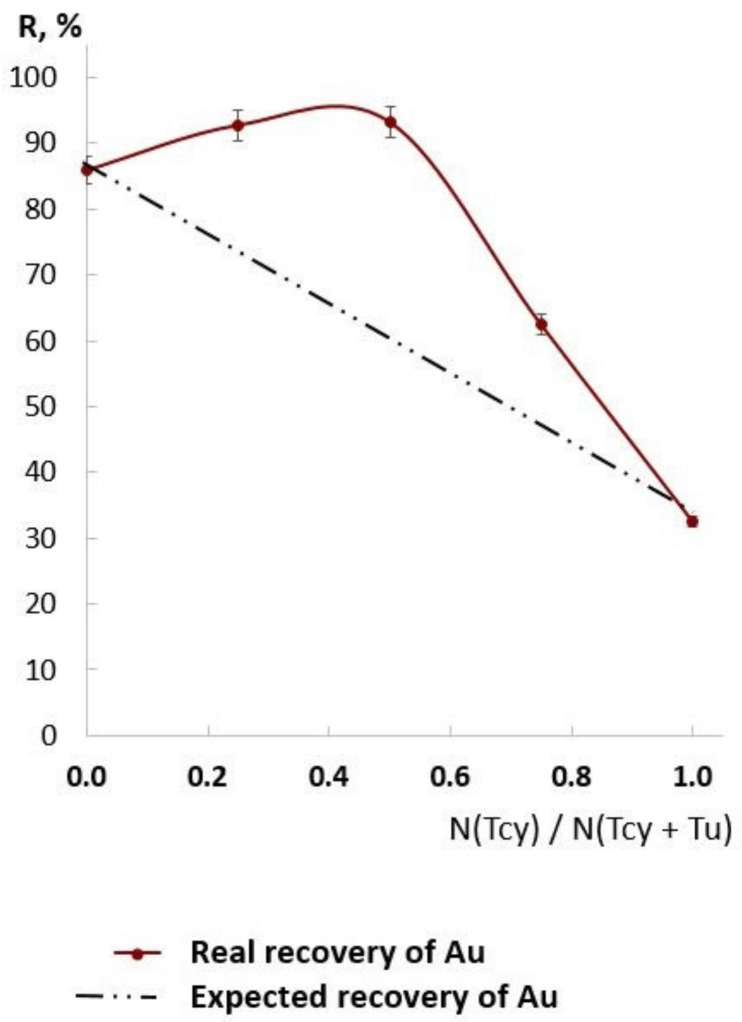
The isomolar series of desorption of Au using thiourea–thiocyanate mixture in 0.2 M HCl (*n* = 3, *p* = 0.95). The sum of molar concentrations of thiourea and thiocyanate was equal to 0.5 M.

**Figure 6 molecules-27-06746-f006:**
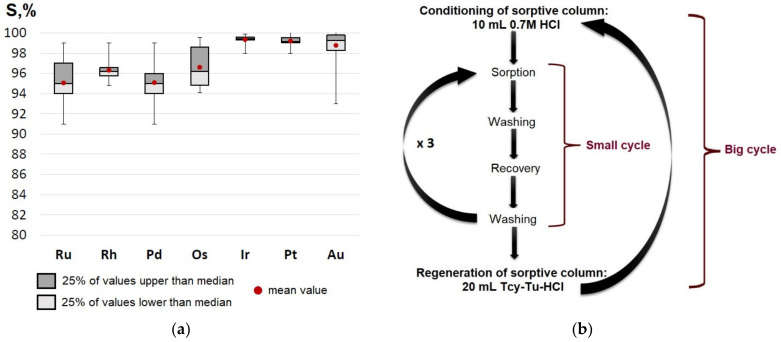
(**a**) Box plot of probability distribution of sorption degrees for 10 sequential big cycles of sorption–recovery. (**b**) Scheme of using the sorbent in small and big cycles.

**Figure 7 molecules-27-06746-f007:**
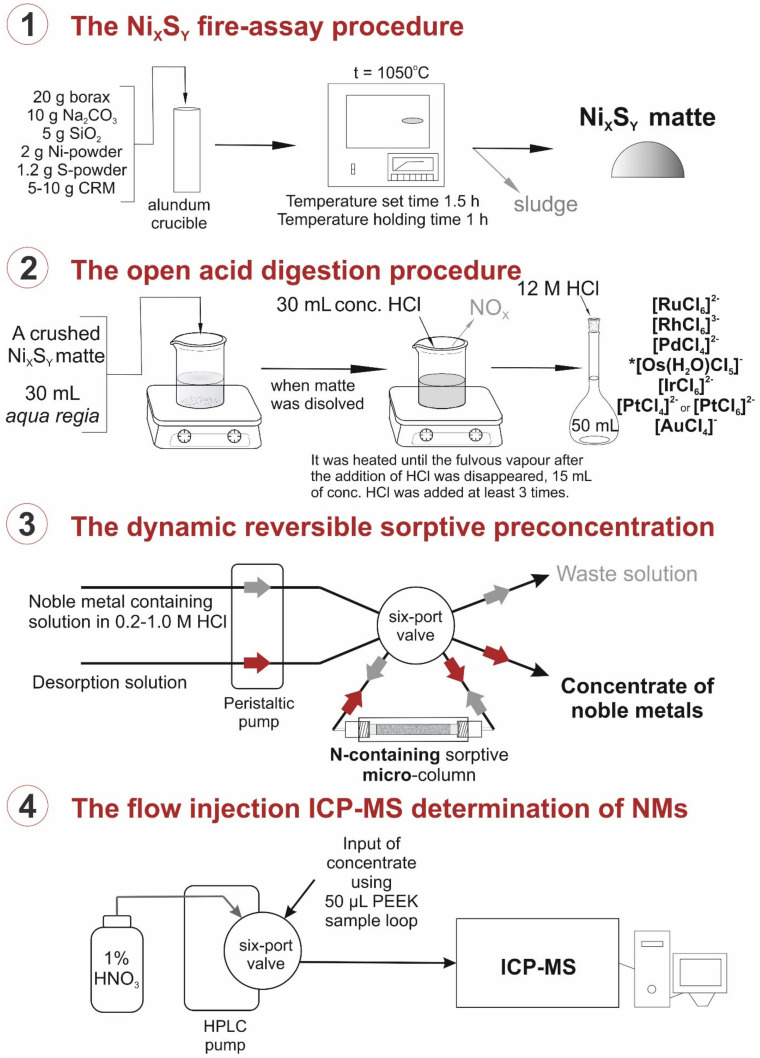
The full scheme of complex procedure of NM determination. * About 30–35% of initial Os was in solution after open acid digestion. Supposedly, Os is present in this complex ionic form.

**Table 1 molecules-27-06746-t001:** Examples of S- and/or N-containing complex-forming resins for NM recovery.

Name of Sorbent	Functional Group	Analytes	Sorptive Capacity, mg g^−1^	NM Sorptive Conditions	NM Recovery in Solution	Determination	Issue
Modified SBA-15 silica	Thiol-	Pd	190	Static mode, >80% in 0.1–1.0 M HNO_3_ or HCl solutions	65% recovery by 1.0 M thiourea solution	GFAAS	[19]
	Amine-	Pd	68	Static mode, >80% in 0.05–0.5 M HNO_3_ or HCl solutions	100% recovery by 0.6 M thiourea solution		
Imidazoline group- containing chelating fiber	Imidazoline group	Ru, Rh, Pd, Os, Ir, Pt, Au	69.1 89.8 180.8 194.9 195.9 184.3 724.0	Dynamic mode in 0.1–1.0 M HCl	Pd, Pt, and Au recovery > 96% by subsequently passing mixture of 1 M HCl and 5% thiourea and then 1 M HClO_4_ and 5% thiourea. No data about desorption of Ru, Rh, Os, or Ir.	ICP-AES	[25]
Aminothioether sorbent	Aminothioether group	Ru, Rh, Pd, Ir, Pt, Au	300 600 3000 550 1400 3600	Static mode in 1–3 M HCl	No data, determination in solid concentrate	ETAAS	[20]
Chemically modified silica gel	Dithiocarbamate group	Ru, Rh, Pd, Os, Ir, Pt, Au	No data 11.3 * 17 * 11.4 * 11.5 * 13.7 * 7.9 *	Two-column procedure: Au, Pd, and Pt at 20 °C in 0.5–4.0 M HCl and Ru, Rh, Ir, and Os at 95 °C in the presence of 0.025 M SnCl_2_	Au, Pd, and Pt were eluted by 10% (*w*/*w*) thiourea solution in 1.0 M HCl at 20⁰C and Ir, Ru, Os, and Rh 10% (*w*/*w*) thiourea solution in 1.0 M HCl at 95 °C	ICP-AES, ICP-MS	[21]
POLYORGS XI	Benzimidazole group	Ru, Rh, Pd, Pt, Au	108 280 300 297 990	Static and dynamic mode (8 cm^3^ wet sorbent) from 2 M HCl	No data, determination in solid concentrate	XRF	[7]
POLYORGS XVII	1,3(5)-Dimethylpyrazol	Rh, Pd, Pt, Au	58 120 144 180	Static and dynamic mode, >80% in 0.05–2 M HCl	Digestion of sorbent in HNO_3_ in a microwave oven	ICP-AES	[8]
POLYORGS IV	3(5)-Methylpyrazole groups	Rh, Pd, Pt, Au	30 100 100 660	Static mode (1 h boiling in 1 M HCl) or dynamic mode.	Elution with 2% thiourea solution in 1 M HCl under microwave heating or with acetone at room temperature	ETAAS	[9,10]
Amberlite IRC-78	Iminodiacetic group	Pd, Pt, Au	58.5 * 60 * 128 *	Static mode, 0.1–4.0 M HCl	>95% recovery by 0.25 M thiourea	ICP-AES	[11,26]
Lewatit TP-214	Thiourea	Pd	172	Static mode, pH 8, 300 min	67% recovery by 0.1 M HCl	AAS	[12]
Hydrazono-imidazoline modified cellulose	Hydrazono-imidazoline group	Pd, Pt, Au	88 105 75	Static mode, pH 4	>98% recovery by 0.1 M HNO_3_ with 0.2 M thiourea	ICP-AES	[16]
Pyridine functionalized TiO2 nanoparticles	Pyridine with tertiary nitrogen	Pd	61	Dynamic mode, pH 7	1 M thiourea in 0.1 M HCl solution	FAAS	[17]
Lewatit MonoPlus TP-220	Pyridine group and tertiary amine group	Pd Pt Au	10.0 9.9 9.9	0.1–6.0 M HCl	Au was quantitatively removed by thiourea and acidic thiourea solutions	AAS	[18]

* Calculated from meq g^−1^ or mmol g^−1^ presented in issue.

**Table 2 molecules-27-06746-t002:** Examples of N-containing ion-exchanger resins for NM recovery.

Name of Sorbent	Functional Group	Analytes	Sorptive Capacity, mg g^−1^	NM Sorptive Conditions	NM Recovery in Solution	Determination	Issue
Dowex 1 × 8(Cl-form)	Quaternary ammonium	Ru, Pd, Ir, Pt, Au	No data	Dynamic mode, 0.5 mL column, 10 mL 0.4 M HCl + Cl_2_	Hot (t = 90 °C) 12 M HNO_3_ recovers > 90% of Pd, Ir, Pt, and Au and ~ 34% of Ru	ICP-MS	[5]
Isolute SAX	Quaternary ammonium	Pd, Pt, Rh	9.6 *, 9.3 *, No data	Dynamic mode, 0.001–1 M HCl	100% Pt and 100% Pd recovered by 1 M thiourea pH 2. Rh was not recovered.	ICP-AES	[6]
Amberlyst A21	Dimethylamine	Pd	100	Static mode, pH 2, 780 min	55–65% recovery by 0.1–4 M KOH or 0.1–0.5 M thiourea	AAS	[12]
Oasis Max	Quaternary ammonium	Rh, Pd, Pt, Au	No data 9.6 12.3 174 *	Dynamic mode, 0.1 M HCl	Pd, Pt, and Au > 95% recovery by 0.5 M thiourea in 1 M HCl	ICP-AES	[13]
Bio-Rad AG 1 × 8	Quaternary ammonium	Ru, Pd, Pt, Ir	No data No data 50 No data	Dynamic mode, HCl solutions	5 M HCl with 5 M HClO_4_	ICP-MS	[14,15]

* Calculated from meq g^−1^ or mmol g^−1^ presented in issue.

**Table 3 molecules-27-06746-t003:** Degrees of NM chlorocomplex sorption (S, %).

NM Complex	Concentration of HCl in Solution, M									
	0.10	0.20	0.40	0.60	1.00	1.25	1.50	2.00	3.00	6.00
[RuCl_6_]^2−^	95	96	96	95	94	95	96	95	87	ND *
[RhCl_6_]^3−^	97	96	96	94	96	77	78	72	46	ND *
[PdCl_4_]^2−^	100	100	100	99	100	100	99	99	99	ND *
[Os(H_2_O)Cl_5_]^−^	94	95	95	94	95	100	100	99	100	ND *
[IrCl_6_]^2−^	100	100	99	99	99	82	82	75	53	ND *
[PtCl_4_]^2−^ and/or [PtCl_6_]^2−^	100	100	99	100	100	100	100	99	100	ND *
[AuCl_4_]^−^	99	100	100	99	100	100	100	99	100	99

* ND means “no data” here.

**Table 4 molecules-27-06746-t004:** Dependence of sorption degree from Ru chlorocomplex forms in different model solutions (*n* = 3, *p* = 0.95).

[Ru_2_O_2_(H_2_O)_2_Cl_6_]^2−^ and [Ru_2_O(H_2_O)_2_Cl_8_]^2−^ and other ^(1)^	[Ru_2_OCl_10_]^4− (2)^	[RuCl_6_]^3− (3)^	[RuCl_6_]^2− (4)^
41 ± 3	81 ± 4	75 ± 5	96 ± 3

^(1)^ Single-element solution for ICP-MS; ^(2)^ Prepared from salt; ^(3)^ SOMB-6 after simple dissolution in aqua regia; ^(4)^ SOMB-6 after NiS fire assay procedure followed by dissolution in aqua regia.

**Table 5 molecules-27-06746-t005:** The experimental sorptive capacity of PVBC-VP sorbent for NMs in 0.2–1.0 M HCl-containing solutions.

NMs	Ru	Rh	Pd	Os	Ir	Pt	Au
**Sorptive capacity,** **mg g^−1^**	not below 12.4	not below 9.1	101	78	not below 20	140	240

**Table 6 molecules-27-06746-t006:** Difference of efficiency between Tcy-Tu mixture, Tu-KCl mixture, and sequential Tu and Tcy solutions, with the sum of concentrations of components equal to 1.0 M and a ratio of 1:1 (all solutions contain 0.5 M HCl acid) passing through the sorptive column (*n* = 3, *p* = 0.95).

NMs	Recovery (R), %		
	10 mL Tcy-Tu Mixture	Sequential Passing of 5 mL Tu and 5 mL Tcy Collected In One Tube	10 mL Tu-KCl Mixture
Ru	86 ± 1	68 ± 1	36 ± 5
Rh	77 ± 7	57 ± 1	33 ± 7
Pd	**92** ± 5	79 ± 2	70 ± 6
Os	70 ± 5	70 ± 7	15 ± 8
Ir	**90** ± 3	79 ± 2	61 ± 9
Pt	**98** ± 5	81 ± 2	**94** ± 5
Au	**100** ± 3	82 ± 4	85 ± 4

**Table 7 molecules-27-06746-t007:** Limits of detection (3σ) of NMs in ng g^−1^ in initial geological sample.

NMs	Ru	Rh	Pd	Os	Ir	Pt	Au
**LOD, ng g^−1^**	2.2	3.7	14.7	3.4	2.4	6.5	7.8

**Table 8 molecules-27-06746-t008:** Results of CRM (GPt-5, GPt-6, and SARM-7) analysis by ICP-MS (sorption conditions: C(HCl) = 1 M, sorbent: PVBC-VP; desorption conditions: 10 mL Tcy:Tu:HCl = 0.5 M:0.5 M:0.8 M; *n* = 10, *p* = 0.95).

**NMs**	**SARM-7**		
	**^1^ Certified, ng g^−1^**	**Found, ng g^−1^**	**Found, %**
Ru	430 ± 57	400 ± 30	93
Rh	240 ± 13	229 ± 17	95
Pd	1530 ± 32	1520 ± 80	99
Os	63 ± 4	^2^ 22 ± 7	**35**
Ir	74 ± 12	72 ± 14	97
Pt	3740 ± 50	3650 ± 70	98
Au	310 ± 15	285 ± 10	92
**NMs**	**GPt-6**		
	** ^1^ ** **Certified, ng g^−1^**	**Found, ng g^−1^**	**Found, %**
Ru	13 ± 1	12 ± 1	97
Rh	24 ± 3	23 ± 4	96
Pd	578 ± 50	530 ± 45	91
Os	15.6 ± 0.2	^2^ 5 ± 1	**30**
Ir	28 ± 7	32 ± 5	91
Pt	410 ± 40	370 ± 30	90
Au	45 ± 10	48 ± 8	107
**NMs**	**GPt-5**		
	** ^1^ ** **Certified, ng g^−1^**	**Found, ng g^−1^**	**Found, %**
Ru	528 ± 91	584	110
Rh	10 ± 2	9.4 ± 2.5	94
Pd	11.3 ± 1.5	14.6	**129**
Os	353 ± 27	^2^ 124	**35**
Ir	136 ± 10	140	103
Pt	20 ± 4	24 ± 5	**118**
Au	–	14 ± 4	–

^1^ Data from [38], ^2^ About 70–80% of Os escaped as OsO_4_ in the process of open acid digestion of the sample.

## Data Availability

Not applicable.

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
