# Peer review of "Reversible Sorptive Preconcentration of Noble Metals Followed by FI-ICP-MS Determination"

_molecules, 2022, doi:10.3390/molecules27196746_

Round 1
Reviewer 1 Report
The manuscript molecules-1922133 “Reversible sorptive preconcentration of noble metals followed by FI-ICP-MS determination” submitted by Yulia A Maksimova and co-workers is devoted to the simultaneous determination of precious metals at a low level in objects of complex composition. This manuscript is under scope of special issue Advances in Detection of Trace Elements by Analytical Spectroscopy. The work is well organized and concisely written and may be published after minor revision.
Comments to the authors
1. The reference to the previous study of the authors [33] is used too often. This source may not be available to readers, and the data presented in the peer-reviewed work is not enough to reproduce the described experiments.
2. The introduction should be supplemented by consideration of various adsorbents with pyridine functional groups and its derivatives proposed for the adsorption of precious metals, as well as attention should be focused on the advantages of the proposed adsorbent in comparison with other adsorbents with pyridine functional groups. Table 2 should be supplemented with information on other adsorbents with pyridine groups.
3. A more detailed description of the sample preparation should be given in the Materials and Methods section, indicating the mass of the CRM sample, the quantity and ratio of the components of the fire assay charge, the volumes of acids for the decomposition of nickel matte, conversion into concentrated HCl medium, the final volume of the solution after decomposition, and the proportion of dilution before adsorption stage.
3. A more detailed description of the sample preparation should be given in the Materials and Methods section, indicating the mass of the CRM sample, the quantity and ratio of the components of the fire assay charge, the volumes of acids for the decomposition of nickel matte, conversion into concentrated HCl medium, the final volume of the solution after decomposition, and the proportion of dilution before adsorption stage.
4. The volume of the model solution that was passed through the column in the study of the degree of adsorption and the concentration of precious metals in it should be indicated.
5. Details of the experiment of adsorption capacity determination should be given. It is not clear from the text of the manuscript: was the adsorption capacity determined under batch or dynamic conditions? To confirm the results obtained, either sorption isotherms or breakthrough curves should be presented.
6. In Table 5, for some elements, the adsorption capacity is indicated as "not below". What does this mean? Why is the exact value of the adsorption capacity not presented?
7. The last sentence of the introduction states that "The aim of our research is selection of reversible sorptive system for NM simultaneous determination." What other systems for the preconcentration of NM were considered besides PVBC-VP? I suppose the authors should clarify the purpose of the study. Why were other resins with pyridine groups not considered?
8. The model and manufacturer of equipment for fire assay is not specified.
9. Equation (3). Why was PF calculated as the ratio of the volume of the initial solution to the volume of the adsorbent, if at the final stage of the analysis the determination of elements was carried out not in the adsorbent phase by any solid-phase method, but in the solution after elution?
10. Why was the 108Pd isotope used to determine palladium?
11. There is no justification for choosing 115In as an internal standard. NM have an ionization potential of about 8-9 eV and In about 5.7 eV.
12. The authors should more clearly indicate the scientific novelty of the present work both in the Introduction section and in the Conclusion.
Reviewer 2 Report
In this work, a reversible sorptive preconcentration method of noble metals followed by FI-ICP-MS determination was developed. This method reports the combined procedure of noble metal (NM) determination, including fire-assay, acid digestion and reversible dynamic sorptive preconcentration using a new PVBC-VP sorbent as micro-column packed, followed by flow-injection ICP-MS. The proposed method was evaluated using certified reference material.
This manuscript is on the scope of the journal and is of board interest. However, the information presented requires minor revision in view of improving some aspects. General comments have been provided below.
- In the abstract it is stated that the recoveries of Os in certified material are 90%, however the results obtained in the validation of the three certified materials studied show that the recovery is around 35%. Could you explain this difference?
- It is known that ICP-AES and ICP-OES describe the same analytical technique, however it would be appropriate, in the table 1, to use one of the two abbreviations in the manuscript.
- page 4, line 66 -68: This paragraph is unclear and would benefit from rephrasing.
- Figure 2. “Absorbance (A.U.)“ instead of “Absorbance”
- Figure 3: “Absorbance (A.U.)” instead of “Absorbanc”
